# Diet Quality in a Weight Gain Prevention Trial of Reproductive Aged Women: A Secondary Analysis of a Cluster Randomized Controlled Trial

**DOI:** 10.3390/nu11010049

**Published:** 2018-12-27

**Authors:** Julie C. Martin, Lisa J. Moran, Helena J. Teede, Sanjeeva Ranasinha, Catherine B. Lombard, Cheryce L. Harrison

**Affiliations:** 1Monash Centre for Health Research and Implementation, School of Public Health and Preventative Medicine, Monash University, Melbourne, VIC 3004, Australia; Julie.C.Martin@monash.edu (J.C.M.); lisa.moran@monash.edu (L.J.M.); helena.teede@monash.edu (H.J.T.); sanjeeva.ranasinha@monash.edu (S.R.); 2Endocrinology and Diabetes Units, Monash Health, Melbourne, VIC 3004, Australia; 3Department of Nutrition and Dietetics, School of Public Health and Preventative Medicine, Monash University, Melbourne, VIC 3004, Australia; catherine.lombard@monash.edu

**Keywords:** diet quality, nutrition, obesity, prevention, lifestyle, intervention, women, rural

## Abstract

Reproductive-aged women are at high risk for obesity development. Limited research exploring weight gain prevention initiatives and associated modifiable risk factors, including diet quality exists. In a secondary analysis of a 12 month, cluster randomized controlled trial for weight gain prevention in reproductive-aged women, we evaluated change in diet quality, macronutrient and micronutrient intake, predictors of change and associations with weight change at follow-up. Forty-one rural towns in Victoria, Australia were randomized to a healthy lifestyle intervention (*n* = 21) or control (*n* = 20). Women aged 18–50, of any body mass index and without conditions known to affect weight, were recruited. Diet quality was assessed by the Dietary Guideline Index (DGI) and energy, macronutrient, and micronutrient intake as well as anthropometrics (weight; kg) were measured at baseline and 12 months. Results were adjusted for group (intervention/control), town cluster, and baseline values of interest. Of 409 women with matched data at baseline and follow-up, 220 women were included for final analysis after accounting for plausible energy intake. At 12 months, diet quality had improved by 6.2% following the intervention, compared to no change observed in the controls (*p* < 0.001). Significant association was found between a change in weight and a change in diet quality score over time β −0.66 (95%CI −1.2, −0.12) *p* = 0.02. The percentage of energy from protein (%) 0.009 (95%CI 0.002, 0.15) *p* = 0.01 and glycemic index −1.2 (95%CI −2.1, −0.24) *p* = 0.02 were also improved following the intervention, compared to the control group. Overall, a low-intensity lifestyle intervention effectively improves diet quality, with associated weight gain preventions, in women of reproductive age.

## 1. Introduction

For the first time, the global prevalence of overweight and obesity exceeds that of those who are underweight [1]. Overall, obesity prevalence is greater in women compared with men (15% versus 11% respectively) [2]. If the current trends in weight gain continue, obesity prevalence is estimated to affect 21% of women and 18% of men by 2025, threatening the likelihood of the World Health Organization (WHO) target to halt the rise in obesity [1]. Weight gain accumulates progressively, with longitudinal studies demonstrating the highest risk in younger women aged 18 to 40 years, who gain more weight annually than older women (649 versus 494 g/year) [3].

Rural-dwelling women are a vulnerable group who are at further risk, with a higher progressive background weight longitudinally (~730 g/year versus ~600 g/year) [3,4], and a higher prevalence of overweight and obesity, in comparison to urban women [5]. The increased risk in rural populations is associated with a higher level of socioeconomic disadvantage [6,7] coupled with reduced access to health care and health promotion initiatives [8], and the higher costs of maintaining a healthy lifestyle [9]. Given the need to improve obesity preventative behaviors in reproductive-aged women including rural women, effective preventive interventions targeting this population are crucial.

Increasing weight in reproductive-aged women is complex; however, in part, this is associated with a reduction in obesity preventative behaviors, including adequate physical activity levels [10,11], reduced sedentary behaviors [12,13], as well as consuming a balanced diet [11,14,15,16] consistent with evidenced-based dietary guidelines for adults [17,18] relative to individual requirements [19]. 

Diet quality is an important modifiable risk factor for obesity prevention, and is inversely associated with weight gain [20,21,22,23], waist circumference [24,25], body mass index (BMI) [25,26], and a reduced risk of associated chronic disease, including diabetes and cardiovascular disease (CVD) [27]. Derived from the Dietary Guideline Index (DGI) and assessed as an overall score, diet quality provides an overview of diet in its entirety, including the assessment of dietary patterns, nutrient intake, and compliance with national dietary guidelines, and is preferable to assessing individual nutrient intake alone [28]. A higher diet quality score infers the increased consumption of a greater variety of healthy foods, including whole-grains, fruits, vegetables, and dairy and lean meats, and less consumption of energy-dense nutrient-poor foods [18].

Yet, despite its demonstrated association with weight gain, there is limited research examining changes in diet quality within obesity preventative strategies in women of reproductive age [29], and a gap exists in our knowledge and understanding of the associated factors to improve diet quality in these settings. Therefore, we aimed to assess the change in diet quality, the predictors of change, and their association with weight change in rural women, as part of a secondary analysis of a large cluster randomized controlled trial (RCT), with demonstrated efficacy in weight gain prevention in women of reproductive age [30,31,32].

## 2. Materials and Methods

### 2.1. Trial Design

This is a secondary analysis of a 12 month, cluster RCT in rural-dwelling women with detailed study methods previously published [32,33]. The intervention HeLP-her was developed to prevent the progressive weight gain observed in women of reproductive age, by promoting a healthy lifestyle and increasing skills in self-management, problem solving, and goal setting through low-intensity behavior change techniques [30,32]. The cluster RCT was set across 41 Victorian towns with a population of 2000 to 10,000, located between 100 and 400 kilometers from the Melbourne central business district [32,33]. This definition of rurality is in line with the Rural Remote and Metropolitan Areas classification [34]. Participants were recruited as clusters according to the town of residence, between September 2012 and April 2013 [31]. Towns were randomized as clusters to intervention (*n* = 21) or control (*n* = 20), with results analyzed at the individual level [31]. All participants provided written informed consent before they participated in the study [30,32]. The study was conducted in accordance with the Declaration of Helsinki, and the protocol was approved on 28 March 2012 by the Monash Health Human Research Ethics Committee (project number 12034B). This trial is registered with the ANZ clinical trial registry ACTRN12612000115831.

### 2.2. Participants

Participants included women aged between 18–50 years of any weight, who lived in rural communities of moderate socio-economic disadvantage, based on the Socio-Economic Index for Areas [35]. The trial was designed to be low intensity, all-inclusive, and community-based to optimize generalizability. Specific exclusion criteria were minimal, and included only conditions or medications known to affect weight, including prescribed weight control medication, bariatric surgery, and breastfeeding of infants under six months, pregnancy, or a serious physical or psychological illness preventing complete participation [31,32].

### 2.3. Intervention

Participants in intervention towns attended one 60 minute face-to-face group session consisting of 8–15 women, delivered by a trained facilitator at schools or other community facilities [31,32]. The intervention aimed to support self-management and healthy behavior changes in women [32,33]. The facilitators had a tertiary qualification in health science, and undertook a one-day training session which covered program theory, practical components, and motivational interviewing techniques [32,33]. The facilitator delivered key messages based upon the 2013 Australian Dietary Guidelines (ADG) and 2013 Physical Activity guidelines for adults [18,36], incorporated as part of a self-management intervention, including goal setting, problem solving, relapse prevention, and self-monitoring, underpinned by the Social Cognitive Theory [31]. Following the session, participants were guided to complete an intervention manual with self-management activities, whilst receiving remote support in the form of monthly text messages and one personalized health coaching phone call at 12 weeks post-group session, to assess compliance, provide support, and to reinforce program objectives [31,32]. To ensure program fidelity, a self-assessment checklist was completed by the facilitator, and verified by an attending research assistant following each session, to reduce potential reporting bias [32,33].

### 2.4. Control

Participants in the control towns received one general health group session, based on the 2013 Australian Dietary and Physical Activity Guidelines [18,36], and were not provided with any additional advice or support during the study [31,32].

### 2.5. Baseline Measures

Pre-specified demographic, health, anthropometrics, and medical history details of each participant were measured and compiled by trained research staff [31,32].

### 2.6. Anthropometrics

Weight was measured on an electronic scale to the nearest 0.1 kg (Tanita WB110AZ), calibrated prior to weighing periods [31,32]. Each participant’s weight was measured in light clothing without shoes, and with an empty bladder [31,32]. Height was measured by using a portable stadiometer to the nearest 0.1 cm (Mentone Education Centre, Melbourne, Australia). Body mass index (BMI) was calculated by dividing each participant’s weight in kilograms (kg) by height in meters squared [31,32]. The WHO BMI classification system was used to categorize participants as normal weight, overweight, and obese [37].

### 2.7. Dietary Intake

Dietary intake data was collected at baseline and at 12 months using a self-administered, semi-quantitative food frequency questionnaire (FFQ) from the Cancer Council of Victoria, for Epidemiological Studies Version 2 [38]. Participants completed questions regarding their usual dietary intake over the previous 12 months by indicating the frequency (never, monthly, weekly, daily, up to three or more times per day) by which they consumed 80 food items including cereal foods, dairy products, meats and fish, fruit and vegetables, and discretionary foods and beverages, including sweets or savory snacks, and alcoholic beverages [38]. These data were analyzed using NUTTAB 95 nutrient data table for use in Australia, version 1995 [39]. The values for the glycemic index (GI) and the glycemic load (GL) of foods was obtained from Foster-Powell, Holt [40]. The GI is a physiological classification of foods according to the foods’ postprandial glycemic effects in comparison to a reference food, either white bread (GI 100) or glucose (GI 143) [40]. The GL is the amount of available carbohydrate in the portion of food consumed, and it is calculated by multiplying the GI of the food by the dietary carbohydrate content [40]. The higher the GI and the GL of a food the greater the expected rise in blood glucose levels [40]. Nutrients were analyzed as mean intakes. The percentages of carbohydrate, total fat, saturated fat, polyunsaturated fat, monounsaturated fat, and protein to the total energy intake were calculated by multiplying each of the macronutrients by its respective kilojoule (kJ) content per gram (carbohydrate and protein 17 kJ/gram, total fat, saturated fat, polyunsaturated fat, and monounsaturated fat 37 kJ/gram [17], then dividing this value by each participant’s total energy intake to give each macronutrients’ density relative to total energy intake.

### 2.8. Dietary Quality

Diet quality was measured and assessed using the Dietary Guideline Index (DGI), an ‘a priori’ scoring method developed for use in the Australian adult population [41]. The DGI was adapted to include the current ADG [18], and recommendations from the 2013 Australian Guide to Healthy Eating (AGHE) [42] with the exception of salt use, fluid intake, and one saturated fat component, which could not be accurately quantified [41]. The adapted tool contained 13 items, including dietary variety, vegetables, fruit, proportion grains, breads and cereals, lean meat, lean meat proportion, dairy, low fat:whole milk, saturated fat, extra foods, alcohol, and added sugars, each representing a dietary guideline. Each component is scored from zero to 10, where zero indicates non-adherence, and 10 indicates adherence to the dietary guidelines [41]. The total diet quality score range was 0–130, with a higher score denoting greater conformity with the ADG, and therefore an overall higher diet quality [41] (Table 1). 

### 2.9. Randomization

Forty-one Australian towns (clusters) were randomized using a computer-generated list for allocation to intervention (*n* = 21) or control (*n* = 20) [31]. The cluster design of the RCT eliminated potential contamination between participants from the same small community [31]. At baseline, researchers were aware of group allocation [31]. Participants were unaware of their group allocation; however, they were aware that they were participating in a healthy lifestyles research program [31]. At the 12 month data collection point, both participants and new field researchers were blinded to group allocation and previous anthropometric measures [31].

### 2.10. Statistics

To strengthen the association between diet and health outcomes, and to reduce the variability in results, energy misreporters were identified and excluded from the final analysis [43]. Each participant’s total energy intake (EI) was divided by their basal metabolic rate (BMR), calculated using the Schofield equation relevant to their age, sex, and weight [44]. Based on prior recommendations, the Goldberg cut off values were applied to identify energy misreporters; <0.9 for low energy reporters, and >2.1 for high energy reporters [45,46]; thereby, adequate energy reporters (EI/BMR 0.9–2.1) were included in the final analysis. Statistical analyses were conducted with Stata Statistical Software: Release 12.1 (StataCorp LLC, College station, Texas, USA) [47], with a two-sided significance level. A statistical analysis plan was followed under the direction of an experienced biostatistician. Baseline characteristics of all participants were analyzed using cluster adjusted *t* test for continuous variables and cluster-adjusted chi square tests for categorical variables. Linear regression was used to assess the difference between groups from baseline to 12 months. Within-group differences over the study period were analyzed by using paired *t*-tests for continuous variables, and chi-square tests for categorical variables adjusted for group allocation (intervention/control), town clustering, and baseline values of interest. The change in all outcome variables (diet quality and its components, macronutrient and micronutrient intake, and the change in weight) were calculated as 12 months minus the baseline data, and were analyzed by using linear regression analysis (continuous outcomes). A logistic random-intercepts model was used for dichotomous outcomes (whole milk: low-fat milk and saturated fat components)) to account for the measurement at two time points (baseline and 12 months). All analyses were adjusted for group allocation (intervention/control), town clustering, and baseline values. Outcomes were adjusted a priori for factors known to influence diet quality, including age (years), BMI, smoking (yes/no/occasionally), employment (full-time, part-time, no paid work), marital status (not married, married), education (no post school qualification, certificate/diploma/apprentice, bachelor degree and above), income level (Australian Dollar AUD 40,000 or less, Australian Dollar AUD 41–64,000, Australian Dollar AUD 65–80,000, Australian Dollar AUD more than 81,000), group status (intervention/control), and town clustering.

## 3. Results

Results are reported according to the CONSORT statement for cluster RCT, see Appendix A. Overall, 649 women were recruited at baseline. The final analysis for this secondary analysis included 220 participants (*n* = 106 control and *n* = 114 intervention) with plausible energy intakes calculated from the FFQ as previously defined, and matched data at baseline and 12 months. The participant CONSORT diagram is presented in Figure 1.

A total of 42 towns (clusters) were randomized and allocated to control (*n* = 21) and intervention (*n* = 21). One town was excluded due to recruitment barriers, so the final analysis included 41 towns (clusters), with a total of 649 women recruited into either the control (*n* = 301) or intervention (*n* = 348) groups. Accounting for participants with incompatible data (*n* = 18) and participants who were unmatched because they did not attend both baseline and 12 month data collection points (*n* = 222), a total of 409 participants was provided for dietary analysis. Diet quality was measured and assessed by the DGI, including macronutrient and micronutrient intake to assess the change in diet quality for the within- and between-group differences in the control and intervention participants at 12 months. The EI/BMR, based on aged and gender, was calculated for each participant. The Goldberg cut-off of <0.9 and >2.1 was used to exclude energy under-reporters (*n* = 158) and over-reporters (*n* = 31) respectively, so adequate energy reporters (*n* = 220) were included in the final analysis with control (*n* = 106) and intervention (*n* = 114) participants.

There were no statistically significant differences in age, BMI, employment status, marital status, education, income or smoking status between intervention and control participants at baseline (Table 2).

Overall, the intervention significantly improved total diet quality, with a between-group difference of 5.8 (95% CI 2.5, 9.1) *p* = 0.001 on adjusted analysis (Table 3). Within groups, the control group had no significant change over time −0.07 (95% CI −2.3, 2.2) *p* = 0.95, compared to the increased total diet quality scores in the intervention group at 12 months 5.5 (95% CI 3.3, 7.8) *p* < 0.001, accounting for a change in diet quality of 6.2%. There were no other significant between-group differences in diet quality components, with the exception of the diet variety score of −0.02 (95% CI −0.04, −0.001) *p* = 0.04, for which there was a reduction in the intervention group, −0.01 (95% CI −0.02, −0.0005) *p* = 0.04, and no change for the control group 0.005 (95% CI −0.006, 0.02) *p* = 0.37.

Change in diet quality was significantly associated with a change in weight on unadjusted β −0.75 (95% CI −1.3, −0.25) *p* = 0.004, and adjusted analysis β −0.66 (95% CI −1.2, −0.12) *p* = 0.02. These findings support our previous findings [31] of a significant change in weight post-intervention, compared to the control group, −0.87 kg (95% CI −1.62kg, −0.13kg) *p* = 0.02, adjusted for baseline values and the clustering effect by town.

On the analysis of nutrient intake, there was a differential effect of the intervention on the change in percentage of energy from protein (%) 0.009 (95% CI 0.002, 0.15) *p* = 0.01 and glycemic index (GI) −1.2 (95% CI −2.1, −0.24) *p* = 0.02 (Table 4). Within groups, the intervention increased the percentage of energy from protein (%) 0.01 (95% CI 0.006, 0.01) *p* < 0.001, with no change in the controls, 0.001 (95% CI −0.003, 0.005) *p* = 0.66. Within groups, the intervention group decreased GI −1.7 (95% CI −2.3, −1.1) *p* < 0.001, with no change in the controls −0.38 (95% CI −0.94, 0.19) *p* = 0.19. There were no other between-group differences for the change in any diet variables.

The association between the change in total diet quality and demographic, and group allocation and study level variables are reported in Table 5. With the exception of the intervention group allocation, which was associated with a significantly higher diet quality score from baseline to 12 months, 5.8 (95% CI 2.5, 9.1) *p* = 0.001, no other variables were predictive of the change in diet quality overall. 

## 4. Discussion

We explored changes in diet quality, macronutrient and micronutrient intakes, and their association with weight change following a 12 month healthy lifestyle program in reproductive-aged rural dwelling women. We report a significant improvement in diet quality, and a significant association between improved diet quality and weight gain prevention, overall. The percentage of energy from protein (%) and GI were also improved in the intervention group compared to the controls. Group allocation was the only significant factor associated with the change in diet quality.

The improvements in percentage of the energy from protein (%) and GI in the intervention relative to controls are consistent with our previous publication, where we reported an increased fruit intake and reduced snack food, takeaway food, and alcohol consumption in the intervention [31]. It is likely that these dietary changes contributed to an overall improved total diet quality score for the intervention group, relative to the controls. Recent research suggests the effect of combined substitutions of healthy food and beverages is more effective at improving diet quality than with targeting a single food group alone [48], which is consistent with our intervention content and design. Our previous publication of the evaluation of this RCT reported that the face-to-face group education sessions and the mixed delivery modes for receiving lifestyle advice (text messages and phone coaching) mediated effective behavior changes in participants [49]. Goal setting, problem solving, and self-management techniques were identified by participants in the RCT to have influenced behavior changes [50]. Our results are in line with previous weight gain prevention trials that have used similar behavior change techniques of similar intensity, resulting in improved diet quality and its components in young men and women [51], and rural community dwellers [52,53]. These results suggest that multicomponent lifestyle interventions tailored to the target population within local communities can result in positive behavioral changes, including dietary change. We further note that the intervention group was the only significant factor associated with the change in diet quality. This finding further emphasizes the strength of the intervention with regards to its utility across a range of demographic characteristics, including equal representation across all BMI, income and education categories, and recruitment from towns of moderate socio-economic status [31].

Our findings of a significant association between change in weight, and a change in diet quality following the intervention, is supported by a systematic review that has previously reported that those who consume healthier diets experience less weight gain in comparison to those who consume unhealthy diets [29]. Moreover, a recent RCT compared changes in diet quality in overweight and obese adults randomized to a basic or enhanced version of a commercial web-based weight loss program, or a wait-list control group for 12 weeks [54]. This study reported a higher change in diet quality for the enhanced group compared to the control (*p* = 0.03) [54]. This change was significantly associated with greater weight loss, with each 1 point increase in diet quality being associated with a 0.1% greater weight loss, equivalent to consuming one new or different food at least once per week [54]. Therefore our results and the prior research demonstrate the potential of incorporating strategies to improve overall diet quality which can assist with weight gain prevention and weight loss among high risk groups. Future research is needed to rigorously evaluate this association, as differences in dietary intake assessment and measurements of outcome variables increases heterogeneity between the studies, potentially weakening associations between diet quality and longitudinal weight change [29].

The strengths of this study include the recruitment of a diverse range of women from existing community groups, and conducting the sessions in local community settings, facilitating peer support and familiarity to enhance participation [31]. The key features of the study design include theoretical-based, personalized, non-prescriptive behavioral change strategies, the use of mixed delivery modes, and the low intensity design, thereby minimizing participant and facilitator burden [31]. Facilitator training, delivery methods, and resources were standardized to ensure high program fidelity [31]. Limitations include self-report FFQs, which can result in misreporting, recall error, measurement error, and induce social desirability bias [55]. However, the FFQ applied in this study is validated against weighed food records with Australian populations, and has been shown to provide a useful method of measuring habitual dietary intake in population settings [56,57], and is generally acceptable as a main method of dietary intake in a study of this type and size. We applied the Goldberg cut-off specific to age and gender to exclude implausible energy reporters, which is likely to increase the plausibility of our findings [46]. Although minimal exclusion criteria was applied at the recruitment stage to optimize generalizability [31], after applying the different levels of data exclusion, including participants with unmatched data and those with implausible energy intakes, the available data for analysis was reduced. Additionally, due to difficulty in accurately interpreting data on parity, we were unable to include this variable in our regression models as a potential confounder. Parity has previously shown to effect diet quality in women [58].

## 5. Conclusions

We report that a low intensity weight gain prevention trial aimed at reproductive aged women, with mixed delivery modes and social support, improved overall diet quality, and is significantly associated with weight gain prevention at 12 months. These findings may help to inform the development of targeted programs and health messages to improve diet quality in women, with the aim of preventing weight gain and reducing risk for ill health and chronic disease.

## Figures and Tables

**Figure 1 nutrients-11-00049-f001:**
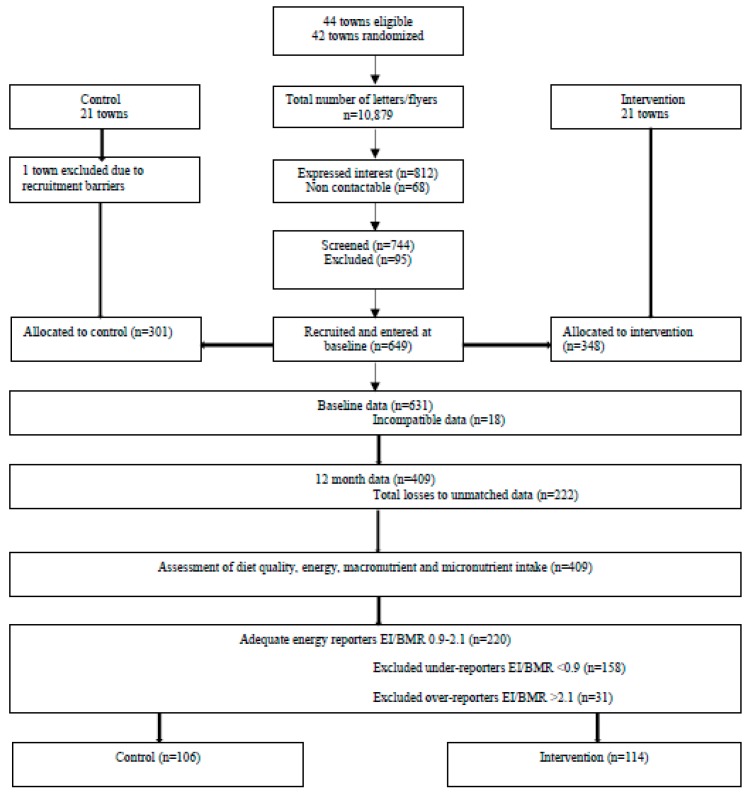
The flow of participants in the final analysis of the HeLP-her rural sub-study. Figure 1 adapted from [31].

**Table 1 nutrients-11-00049-t001:** Modified version of the DGI.

2013 Australian Dietary Guidelines	DGI Component and Description	Maximum Score (10)	Intermediate Score (5)	No (0)
Enjoy a wide variety of nutritious foods	Dietary variety: proportions of foods for each core food group that were consumed at least once per week	100%	50%	0%
Eat plenty of vegetables, legumes and fruits	Vegetables: servings of vegetables and legumes per day	≥5	2.5	0
	Fruit: servings of fruit per day	≥2	1	0
Eat plenty of cereals (including breads, rice, pasta, and noodles), preferably whole-grain	Cereals: frequency of consumption of breads and cereals per day	≥6	3	0
	Wholegrain cereals: proportion of whole-meal/wholegrain bread consumed relative to total bread	100%	50%	0%
Include lean meat, fish, poultry or alternatives	Meat and meat alternatives: frequency of consumption of lean meats and alternatives per day	≥2.5	1.25	0
	Lean protein sources: proportion of lean meats and alternative relative to total meats and alternatives	100%	50%	0%
Include milks, yoghurts, cheeses, and/or alternatives Reduced fat varieties should be chosen, where possible	Dairy foods: frequency of consumption of dairy products per day	≥2.5	1.25	0
	Low fat/reduced fat dairy: type of milk usually consumed	Low-fat milk	N/A	Whole milk
Limit saturated fat intake and moderate total fat intake	Saturated fat intake: type of milk usually consumed	Low-fat milk		Whole milk
Limit your alcohol intake if you choose to drink	Alcohol: frequency of consumption of all alcoholic beverages per day	≤1	1.5	≥2
Consume only moderate amounts of sugars and foods containing added sugars	Added sugars: frequency of consumption of soft drink, cordial, fruit juice, jam, chocolate, confectionary per day	F < 1.25	1.25	F > 1.25
Prevent weight gain: by being physically active and eating according to your energy needs	Extra foods: frequency of consumption of extra foods per day	F < 2.5	2.5	F > 2.5
	Total DGI score	0–130		

**Table 2 nutrients-11-00049-t002:** Baseline characteristics of participants.

Characteristic	Control *n* = 106	Intervention *n* = 114	*p*-Value
Age (years)	39.9 ± 6.2	40.9 ± 5.3	0.22
Body mass index (BMI) (kg/m^2^)	26.7 ± 5.1	27.4 ± 6.1	0.4
Employment			0.65
Full time	14 (13.2)	20(17.7)	
Part time	70 (66.0)	68 (60.2)	
No paid work	22 (20.8)	25 (22.1)	
Marital status			0.05
Not married	15 (14.2)	7 (6.1)	
Married	91 (85.9)	107 (93.9)	
Education			0.19
No post school qualification	18 (17.1)	12 (10.5)	
Certificate/Diploma/Apprentice	49 (46.7)	49 (43.0)	
Bachelor degree and above	38 (36.2)	53 (46.5)	
Income			0.37
Australian dollar AUD 40,000 or less	18 (17.5)	17 (15.7)	
Australian dollar AUD 41–64,000	23 (22.3)	19 (17.6)	
Australian dollar AUD 65–80,000	26 (25.2)	20 (18.5)	
More than Australian dollar AUD 81,000	36 (35.0)	52 (48.2)	
Smoking			0.3
No	97 (91.5)	105 (93.8)	
Yes	2 (1.9)	4 (3.6)	
Occasionally	7 (6.6)	3 (2.7)	

Data are presented as mean ± SD for continuous variables or the frequency and percentage for categorical variables, and were analyzed using cluster-adjusted *t*-test for continuous variables and cluster-adjusted chi-square tests for categorical variables.

**Table 3 nutrients-11-00049-t003:** Change in diet quality from baseline to 12 months between control and intervention participants.

Diet Quality	Control *n* = 106	Percentage Change %	Intervention *n* = 114	Percentage Change %	Unadjusted Difference β (95% Confidence Interval CI) *p*-Value	Adjusted Difference ^1^ β (95% Confidence Interval CI) *p*-Value
Total diet quality						
Baseline	84.2 (17.6)		83.0 (16.6)			
Follow-up	84.2 (17.2)		88.5 (17.5)			
Mean change (95% CI) *p*-Value	−0.07 (−2.3, 2.2) 0.95	0	5.5 (3.3, 7.8) <0.001	6.2	5.6 (2.2, 9.0) 0.002	5.8 (2.5, 9.1) 0.001
Dietary variety total						
Baseline	0.66 (0.09)		0.66 (0.09)			
Follow-up	0.67 (0.09)		0.65 (0.09)			
Mean change (95% CI) *p*-Value	0.005 (−0.006, 0.02) 0.37	1.5	−0.01(−0.02, 0.0005) 0.04	−1.5	−0.02 (−0.04, 0.004) 0.10	−0.02 (−0.04, −0.001) 0.04
Vegetable total						
Baseline	2.4 (1.1)		2.5 (1.1)			
Follow-up	2.4 (1.1)		2.4 (0.85)			
Mean change (95% CI) *p*-Value	0.02 (−0.12, 0.15) 0.80	0	−0.07 (−0.24, 0.10) 0.40	−4.2	−0.09 (−0.28, 0.10) 0.34	−0.08(−0.26, 0.11) 0.41
Fruit total						
Baseline	1.6 (0.90)		1.7 (1.1)			
Follow-up	1.6 (0.92)		1.7 (1.0)			
Mean change (95% CI) *p*-Value	−0.004 (−0.14, 0.13) 0.95	0	−0.009 (−0.17, 0.15) 0.91	0	−0.005 (−0.20, 0.20) 0.96	0.04(−0.17, 0.25) 0.71
Proportion grains						
Baseline	0.67 (0.58, 0.76)		0.74 (0.44)			
Follow-up	0.69 (0.46)		0.83 (0.38)			
Mean change (95% CI) *p*-Value	0.02 (−0.04, 0.09) 0.49	2.9	0.09 (0.02, 0.16) 0.02	10.8	−0.0009 (−0.03, 0.03) 0.94	−0.002 (−0.03, 0.03) 0.89
Breads and cereals total						
Baseline	4.1 (1.6)		4.4 (1.4)			
Follow-up	3.9 (1.5)		4.2 (1.5)			
Mean change (95% CI) *p*-Value	−0.22 (−0.47, 0.03) 0.08	−5.1	−0.28 (−0.59, 0.03) 0.08	−4.8	−0.06 (−0.40, 0.27) 0.70	−0.06 (−0.40, 0.28) 0.72
Lean meat total						
Baseline	2.3 (1.1)		2.4 (0.95)			
Follow-up	2.3 (1.3)		2.5 (1.0)			
Mean change (95% CI) *p*-Value	−0.006 (−0.20, 0.19) 0.95	0	0.11 (−0.07, 0.29) 0.22	4	0.12 (−0.14, 0.38) 0.37	0.15 (−0.12, 0.42) 0.26
Lean meat proportion						
Baseline	0.82 (0.11)		0.82 (0.09)			
Follow-up	0.84 (0.09)		0.84 (0.08)			
Mean change (95% CI) *p*-Value	0.02 (0.0002, 0.04) 0.05	2.4	0.02 (0.005, 0.04) 0.01	2.4	0.0009 (−0.03, 0.03) 0.94	0.002(−0.03, 0.03) 0.89
Dairy total						
Baseline	1.9 (0.81)		1.8 (0.67)			
Follow-up	1.8 (0.68)		1.8 (0.71)			
Change mean (95% CI) *p*-Value	−0.13 (−0.26, 0.005) 0.06	−5.6	0.05 (−0.07, 0.18) 0.42	0	0.18 (0.02, 0.35) 0.03	0.14 (−0.06, 0.34) 0.16
Extra foods total ^2^						
Baseline	4.7 (2.0)		5.0 (2.3)			
Follow-up	4.3 (1.9)		4.1 (2.0)			
Mean change (95% CI) *p*-Value	−0.40 (−0.74, −0.06) 0.02	−9.3	−0.84 (−1.2, −0.48) <0.001	−22.0	−0.44 (−0.97, 0.10) 0.11	−0.36(−0.96, 0.24) 0.24

Data are presented as mean ± SD, and analyzed using paired *t*-test for the within-group change. Linear regression was used to assess the difference between groups over the length of the study, and are presented as beta, 95% confidence intervals, and *p*-value. Percentage change was calculated by dividing the baseline mean value by the follow-up mean value minus one, then multiplied by one hundred. All data were adjusted for group (intervention/control) and town cluster. ^1^ Additional adjustment for age, BMI, smoking, working status, marital status, qualifications, income level, group status (intervention/control), and town clustering. ^2^ Extra foods total includes the alcohol and the added sugars components of the DGI; therefore, the results for alcohol and the added sugars components of the DGI are not shown separately in this table. Notes: The saturated fat and whole milk:low-fat components of the DGI were analyzed by logistic random-intercepts model because the outcomes are dichotomous and are measured at two time points (baseline and 12 months). On unadjusted analysis (0.03 95% CI −2.7 to 2.8 *p* = 0.981) (one result applies to both variables because both variables score the same foods in the same way), these diet quality components were not significantly different between the intervention and control.

**Table 4 nutrients-11-00049-t004:** Change in energy, macronutrient, and micronutrient intake from baseline to 12 months between the control and intervention participants.

Energy and Nutrients	Control *n* = 106	Intervention *n* = 114	Unadjusted Difference β (95% Confidence Interval CI) *p*-Value	Adjusted Difference ^1^ β (95% Confidence Interval CI) *p*-Value
Energy (kJ)				
Baseline	8051.7 (1827.6)	8286.6 (1990.5)		
Follow-up	7606.5 (1717.2)	7760.3 (1621.8)		
Mean change (95% CI) *p*-value	−445.3(−746.6, −143.9) 0.004	−526.3(−848.9, −203.8) 0.002	−81.1 (−443.2, 281.1) 0.65	−42.9 (−460.0, 374.2) 0.84
Protein (g)				
Baseline	94.3 (26.1)	93.0 (23.6)		
Follow-up	89.5 (26.2)	91.8 (22.0)		
Mean change (95% CI) *p*-value	−4.8 (−8.6, −0.94) 0.02	−1.2 (−5.2, 2.7) 0.54	3.5 (−1.4, 8.5) 0.16	3.5 (−1.7, 8.8) 0.18
% protein				
Baseline	0.20 (0.03)	0.19 (0.02)		
Follow-up	0.20 (0.03)	0.20 (0.03)		
Mean change (95% CI) *p*-value	0.001 (−0.003, 0.005) 0.66	0.01 (0.006, 0.01) <0.001	0.009 (0.003, 0.01) 0.003	0.009 (0.002, 0.15) 0.01
CHO (g)				
Baseline	191.3 (51.2)	198.4 (51.2)		
Follow-up	180.6 (45.2)	182.5 (42.7)		
Mean change (95% CI) *p*-value	−10.7 (−19.0, −2.4) 0.01	−15.9 (−25.4, −6.4) 0.001	−5.2 (−16.6, 6.1) 0.36	−3.3 (−15.3, 8.7) 0.58
% CHO				
Baseline	0.40 (0.06)	0.41 (0.06)		
Follow-up	0.40 (0.05)	0.40 (0.06)		
Mean change (95% CI) *p*-value	0.0009 (−0.008, 0.01) 0.85	−0.006 (−0.02, 0.003) 0.20	−0.007(−0.02, 0.007) 0.31	−0.005(−0.02, 0.009) 0.46
Fat (g)				
Baseline	80.6 (21.8)	82.6 (24.7)		
Follow-up	76.6 (21.2)	77.1 (21.5)		
Mean change (95% CI) *p*-value	−3.9 (−7.7, −0.18) 0.04	−5.5 (−9.3, −1.8) 0.004	−1.6 (−6.3, 3.1) 0.49	−1.6 (−7.5, 4.3) 0.59
% Fat				
Baseline	0.37 (0.04)	0.37 (0.05)		
Follow-up	0.37 (0.04)	0.36 (0.05)		
Mean change (95% CI) *p*-value	0.002 (−0.005, 0.009) 0.59	−0.002 (−0.01, 0.006) 0.64	−0.004 (−0.01, 0.006) 0.45	−0.005 (−0.02, 0.008) 0.42
SFA (g)				
Baseline	34.4 (11.0)	34.5 (11.9)		
Follow-up	32.3 (10.3)	31.6 (10.2)		
Mean change (95% CI) *p*-value	−2.1 (−3.8, −0.34) 0.02	−2.8 (−4.5, −1.1) 0.001	−0.78 (−3.0, 1.4) 0.48	−0.83 (−3.4, 1.8) 0.52
% SFA				
Baseline	0.16 (0.03)	0.15 (0.03)		
Follow-up	0.16 (0.03)	0.15 (0.03)		
Mean change (95% CI) *p*-value	−0.0004 (−0.004, 0.004) 0.83	−0.003 (−0.008, 0.002) 0.24	−0.002 (−0.008, 0.003) 0.40	−0.003(−0.01, 0.004) 0.41
MUFA (g)				
Baseline	28.6 (7.7)	29.6 (9.2)		
Follow-up	27.2 (7.7)	28.0 (8.2)		
Mean change (95% CI) *p*-value	−1.4 (−2.8, −0.03) 0.04	−1.7 (−3.1, −0.26) 0.02	−0.28 (−2.1, 1.6) 0.76	−0.17(−2.5, 2.1) 0.88
% MUFA				
Baseline	0.13 (0.02)	0.13 (0.02)		
Follow-up	0.13 (0.02)	0.13 (0.02)		
Mean change (95% CI) *p*-value	0.0005 (−0.003, 0.004) 0.76	0.0007 (−0.003, 0.004) 0.66	0.0003 (−0.005, 0.005) 0.91	0.00002(−0.006, 0.006) 0.10
PUFA (g)				
Baseline	10.6 (3.9)	11.4 (4.0)		
Follow-up	10.4 (3.8)	10.8 (3.9)		
Mean change (95% CI) *p*-value	−0.16 (−0.80, 0.48) 0.62	−0.60 (−1.3, 0.13) 0.11	−0.44 (−1.4, 0.56) 0.38	−0.46 (−1.6, 0.67) 0.41
% PUFA				
Baseline	0.05 (0.02)	0.05 (0.01)		
Follow-up	0.05 (0.01)	0.05 (0.01)		
Mean change (95% CI) *p*-value	0.002 (−0.0006, 0.004) 0.16	0.0001 (−0.002, 0.003) 0.92	−0.001 (−0.006, 0.003) 0.47	−0.002 (−0.006, 0.002) 0.31
Fiber (g)				
Baseline	21.5 (5.5)	22.9 (6.7)		
Follow-up	21.1 (5.6)	22.6 (6.0)		
Mean change (95% CI) *p*-value	−0.35 (−1.2, 0.51) 0.42	−0.29 (−1.5, 0.90) 0.63	0.07 (−1.2, 1.3) 0.91	0.32 (−1.1, 1.7) 0.65
Cholesterol (mg)				
Baseline	314.8 (98.8)	315.0 (111.2)		
Follow-up	301.0 (102.7)	310.0 (113.1)		
Mean change (95% CI) *p*-value	−13.8 (−30.7, 3.1) 0.11	−5.0 (−23.0, 13.0) 0.58	8.8 (−14.3, 31.9) 0.45	8.4 (−16.7, 33.6) 0.50
GI				
Baseline	50.7 (3.6)	51.2 (4.0)		
Follow-up	50.4 (3.5)	49.5 (3.7)		
Mean change (95% CI) *p*-value	−0.38 (−0.94, 0.19) 0.19	−1.7 (−2.3, −1.1) <0.001	−1.3 (−2.1, −0.56) 0.001	−1.2 (−2.1, −0.24) 0.02
GL				
Baseline	96.8 (29.8)	101.2 (29.6)		
Follow-up	90.6 (25.7)	90.2 (25.2)		
Mean change (95% CI) *p*-value	−6.2 (−11.0, −1.3) 0.01	−10.9 (−16.3, −5.5) <0.001	−4.8 (−11.3, 1.7) 0.15	−3.4 (−10.4, 3.6) 0.33
Calcium (mg)				
Baseline	970.0 (294.6)	929.3 (248.0)		
Follow-up	919.4 (259.2)	936.4 (245.1)		
Mean change (95% CI) *p*-value	−50.6 (−96.3, −4.8) 0.03	7.1 (−34.7, 48.9) 0.74	57.6 (8.0, 107.3) 0.02	48.3 (−11.6, 108.1) 0.11
Iron (mg)				
Baseline	13.5 (3.6)	14.0 (4.4)		
Follow-up	12.8 (3.4)	13.7 (3.8)		
Mean change (95% CI) *p*-value	−0.69 (−1.3, −0.11) 0.02	−0.35 (−1.2, 0.50) 0.42	0.33 (−0.56, 1.2) 0.46	0.49 (−0.46, 1.4) 0.31
Folate (µg)				
Baseline	265.9 (67.4)	276.5 (91.1)		
Follow-up	258.2 (60.4)	268.5 (69.9)		
Mean change (95% CI) *p*-value	−7.7 (−18.3, 2.9) 0.15	−8.0 (−25.0, 9.1) 0.36	−0.28 (−16.3, 15.7) 0.97	4.8 (−12.1, 21.8) 0.57
Sodium (mg)				
Baseline	2557.1 (713.1)	2605.9 (755.4)		
Follow-up	2379.6 (604.8)	2421.7 (638.7)		
Mean change (95% CI) *p*-value	−177.5 (−287.0, −68.0) 0.002	−184.2 (-300.8, −67.7) 0.002	−6.7 (−140.7, 127.2) 0.92	−4.9 (−159.2, 149.3) 0.95

Data are presented as mean ± SD and analyzed using paired Student’s *t*-tests for the within-group change. Linear regression were used to assess the difference between groups over the length of the study, and are presented as beta, 95% confidence interval, and *p*-value. All data were adjusted for group (intervention/control) and town cluster. ^1^ Additional adjustment for age, BMI, smoking, working status, marital status, qualifications, income level, group status (intervention/control), and town clustering.

**Table 5 nutrients-11-00049-t005:** Association between baseline demographic, anthropometric, and study-level factors to the change in total diet quality from baseline to 12 months.

Characteristics	Unadjusted β (95% Confidence Interval CI) *p*-Value	Adjusted ^1^ β (95%ConfidenceInterval CI) *p*-Value
Age (years)	0.14 (−0.09, 0.37) 0.23	0.18 (−0.14, 0.50) 0.25
BMI (kg/m^2^)	0.12 (−0.24, 0.47) 0.51	0.12 (−0.28, 0.53) 0.55
Smoking		
No	Ref	Ref
Yes	−0.15(−4.8, 4.5) 0.95	−0.72(−6.4, 4.9) 0.80
Occasionally	4.0 (−2.2, 10.2) 0.20	4.7 (−0.85, 10.2) 0.10
Employment		
Full time	Ref	Ref
Part time	0.99 (−3.9, 5.9) 0.69	4.0 (−1.0, 9.0) 0.12
No paid work	0.94 (−4.1, 6.0) 0.71	3.7 (−3.0, 10.4) 0.27
Marital status		
Not married	Ref	Ref
Married	2.1 (−2.2, 6.3) 0.33	1.8 (−2.5, 6.1) 0.40
Education		
No post school qualification	Ref	
Certificate/diploma/apprentice	4.1 (−0.41, 8.5) 0.07	3.8 (−1.4, 9.0) 0.15
Bachelor degree and above	2.5 (−2.5, 7.5) 0.32	2.3 (−3.5, 8.0) 0.43
Income		
Australian dollar AUD 40,000 or less	Ref	Ref
Australian dollar AUD 1–64,000	0.61 (−4.5, 5.7) 0.81	0.62(−4.5, 5.7) 0.81
Australian dollar AUD 65–80,000	−0.32 (−5.7, 5.1) 0.91	−0.72 (−7.1, 5.7) 0.82
More than Australian dollar AUD 81,000	0.31 (−5.0, 5.6) 0.91	0.20 (−5.8, 6.2) 0.95
Group		
Control	Ref	Ref
Intervention	5.6 (2.2, 9.0) 0.002	5.8 (2.5, 9.1) 0.001

Data are presented as β (95% confidence interval, CI) *p*-value, and were analyzed using linear regression adjusted for group allocation (intervention/control) and town clustering. ^1^ Additional adjustment for age, BMI, smoking, working status, marital status, qualifications, income level, group status (intervention/control), and town clustering.

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
