# Peer review of "Diet Quality in a Weight Gain Prevention Trial of Reproductive Aged Women: A Secondary Analysis of a Cluster Randomized Controlled Trial"

_nutrients, 2018, doi:10.3390/nu11010049_

Round 1
Reviewer 1 Report
The authors report a secondary analysis of data collected in the HeLP-her study aiming to assess the association between the change in diet quality and weight change during the one year study. These data suggest that diet quality is associated weight change in this cohort of reproductive age women. The paper requires revisions to the method section to clarify the study design and attrition rate/ data exclusions that were made. The paper would also benefit from editing to ensure clarity. Specific comments are as follows:
Abstract
Abstract implies that 649 women were included in the analysis, whereas in the result section it is reported that only 220 participants were included in the data analysis. Please revise.
Identify -0.66 as beta coefficient.
Clarify units of reported statistics in abstract (e.g. diet quality, protein density).
Line 30-31: this sentence needs revising.
Introduction
Clarify wording of lines 44-47. Current sentence structure indicates women are “less likely to partake in obesity preventative behaviors including…consumption of a high energy intake”
Reword “high energy intake (>= 11,200 kJ/d)” (line 47). “High” is a relative term. The real concern is not high energy intake but rather energy consumed in excess of need. Also, there is not a cut point for excess energy since there is such heterogeneity in energy requirements, remove reference to a specific kj cutpoint.
Revise point about “consuming a balanced diet” to describe current dietary guidelines.
Revise lines 61-63 to clarify logic. Why are weight gain prevention programs that improve diet quality, specifically, preferable? Long-term health benefits?
Methods
Line 78: Please clarify randomizing towns, but analyzing at the individual level. Consort diagram suggests randomization of both towns and participants. Your previous paper states that there was only one level of randomization at the town level. Please revise your CONSORT diagram so that it is clear that randomization only occurred once.
Line 99: Please specify the year of publication of the Australian Dietary Guidelines
Line 138: Remove “validated” and replace with “developed”. You have modified this tool and so it is not longer accurate to infer this is a validated index.
Line 156: Please provide more details about randomization (what statistical program, etc.?)
Line 177: Between group differences should not be tested using a paired student’s t-test.
Results
Figure 1: In addition to the changes required to this figure to clarify the randomization, the total losses to unmatched data (n=222) need to be described. Why were these data unmatched? Suggest you change the format to reflect the figure in your previous publication (reference 18).
Table 2: remove “all” from the title since the characteristics in this table only represent the subgroup included in these analyses. In the table legend change “data is” to “data are”.
Line 214-216: For clarity this sentence needs revising. Also change “In our previous study” to “in our previous publication” since you are referring to the same study.
Throughout the results section, you need to state what the values are and add units where appropriate.
Line 230: How were protein density and GI calculated? This needs to be added to the methods section.
Table 3: Please indent baseline, follow-up, and mean change in the first column to help the reader. Also bold significant p values.
Discussion: Overall this section should be reduced in length
Lines 263-265: This seems to be an overstatement as the only significant macronutrient changes was in percent protein. Please specify instead of generalizing to all 3 macronutrients.
Lines 279-282: This sentence is unclear, please rephrase.
Line 295 and lines 300-315: Some of this has been previously stated in the discussion; condense.
Line 300: replace “efficacy” this is not an efficacy study.
Line 318-319: You state you data are generalizable because of the few exclusion criteria, however only one third of your original sample was included in the data analysis after all the different levels of data exclusion, which limits the generalizability for the results. This needs to be acknowledged.
Conclusion
Lines 332 and 333 can be combined; the conclusion should state that a low intensity intervention with mixed delivery modes and social support was effective at preventing weight gain so as not to overstate the results.
Line 334: This line states there were 3 large-scale studies, is this a typo? It is unclear where the other 2 studies took place.
Author Response
Please find attached Word document with our responses to reviewer 1.

Reviewer 2 Report
my opinion is that the manuscript is well presented, clear and present conclusions in line with the results.
the introductory section, however, would need an improvement: it should be increased with more information about the factors that modify the weight, above all, attention should be placed on the basal metabolism and energy expenditure, as also a high quality food, if overdosed, leads to constant weight increase
Author Response
Please find attached a Word document of our responses to reviewer 2. Please refer to page 7.

Round 2
Reviewer 1 Report
The authors have significantly improved the manuscript and I have no major concerns. Some minor corrections required:
Minor corrections:
- For ease of interpretation and consistency replace “protein density” with percentage of energy from protein (%) throughout the paper.
-The authors did not include how GI was calculated. Since GI is applied to foods it is unclear what the meaning of an overall GI value is, this needs to be explained since this is one of your findings.
- in Figure 1 you state “entered baseline and 12 month data = 631”), however since you only had 409 women with matched baseline and 12 months data, this implies the 631 figure is incorrect. Suggest you have a value for baseline (n=631) and a value for 12 months (n=409).
Author Response
Comments to reviewer's comments round 2 attached below.
